# Development of the PD9-9 Monoclonal Antibody for Identifying Porcine Bone Marrow-Derived Dendritic Cells

**DOI:** 10.3390/life14091054

**Published:** 2024-08-23

**Authors:** Sang Eun Kim, Young Kyu Kim, Keon Bong Oh, Jeong Ho Hwang

**Affiliations:** 1Department of Stem Cell and Regenerative Biotechnology, KU Institute of Science and Technology (KIT), Konkuk University, Seoul 05029, Republic of Korea; 2Animal Biotechnology Division, National Institute of Animal Science, Rural Development Administration, Wanju-gun 55365, Republic of Korea; sesesese@korea.kr; 3Animal Model Research Group, Korea Institute of Toxicology, 30 Baekhak1-gil, Jeongeup-si 56212, Republic of Korea; youngkyu.kim@kitox.re.kr

**Keywords:** porcine dendritic cells, bone marrow-derived dendritic cells, monoclonal antibody

## Abstract

The purpose of this study was to develop a monoclonal antibody (mAb) that can identify porcine dendritic cells (DCs) that have differentiated from bone marrow progenitor cells. Hybridoma technology was used to obtain mAbs, and bone marrow-derived DCs (BMDCs) were employed as immunogens for producing antibodies. The generated PD9-9 mAbs exhibited considerable reactivity towards porcine BMDCs with applications in flow cytometry and immunostaining. The antibody was composed of heavy immunoglobulin gamma-1 chains and light kappa chains. The PD9-9 mAb recognized fully differentiated porcine BMDCs and cells undergoing DC differentiation. In contrast, bone marrow cells and macrophages were not recognized by PD9-9. In addition, the PD9-9 mAb promoted porcine DC proliferation. Consequently, the PD9-9 mAb may be a biomarker for porcine DCs and will be advantageous for investigating porcine DC biology.

## 1. Introduction

Pigs are economically essential livestock and crucial large animal models for biomedical research, such as xenotransplantation and influenza infection biology [1,2,3,4,5]. Viral infections in pigs result in substantial economic losses and pose a global threat to the pig industry [6,7,8,9]. Major swine viruses, such as porcine reproductive and respiratory syndrome virus, foot-and-mouth disease virus, African swine fever virus, classical swine fever virus, and porcine circovirus type 2, interact with dendritic cells (DCs) throughout viral infection [10,11,12,13]. Therefore, DCs must be investigated because they play a critical role in primary viral infections. DCs are sentinels of the immune system and are professional antigen-presenting cells that frequently reside in tissues near pathogen entry sites [14,15,16,17]. However, the low frequency of DCs in pig tissues and other species complicates their isolation [18,19,20]. Hence, producing DCs in vitro could be a practical substitute.

DCs in mice and humans exhibit a notable level of CD11c expression; thus, CD11c is frequently used as a DC identification marker [14,21,22]. However, because porcine CD11c is also strongly expressed in monocytes and macrophages, it cannot be used as a marker to distinguish porcine DCs from other cell types [23]. In contrast, the Flt3 ligand, which is essential for DC development, can be a reliable porcine DC marker [24]. Flt3 was abundantly expressed in DC precursors and bone marrow-derived DCs (BMDCs) differentiated via Flt3 ligand stimulation, whereas it was undetectable in monocyte-derived DCs or BMDCs differentiated via GM-CSF supplementation [25]. Therefore, evaluating the expression of surface markers, such as MHC II, CD1, CD172a, and CD16, is essential to distinguish GM-CSF–BMDCs from bone marrow precursor cells or macrophages [26,27]. Porcine DC-pathogen interactions have been extensively studied using DCs differentiated in vitro from bone marrow cells (BMCs) using the granulocyte–macrophage colony-stimulating factor (GM-CSF) [28]. Therefore, this study aimed to develop monoclonal antibodies (mAbs) capable of identifying porcine BMDCs differentiated using the GM-CSF.

## 2. Materials and Methods

### 2.1. Animals

Animal experiments were conducted in accordance with the guidelines for the care and use of laboratory animals established by the National Institute of Health. Approval was obtained from the Institutional Animal Care and Use Committee (IACUC) of Konkuk University (approval no. KU15183 and KU16168) before the study. Gnotobiotic miniature pigs were housed in an absolute barrier-containing facility at the Bio-Organ Research Center of Konkuk University (Seoul, Republic of Korea) [29]. BMCs were isolated from stillborn piglets. A single 10-week-old male BALB/c mouse was used to generate hybridomas. The mouse was housed in the Animal Facility of the Department of Animal Biotechnology.

### 2.2. Cells

BMCs were isolated from the humerus, tibia, and femur of stillborn piglets to induce the differentiation of BMCs into DCs. BMDCs were generated as previously described [30]. Non-adherent cells were used for immunization and hybridoma screening.

3D4/2 cells and pulmonary alveolar macrophages were cultured in RPMI-1640 medium (Gibco, Carlsbad, CA, USA) supplemented with 10% heat-inactivated fetal bovine serum (Gibco), 100 U/mL penicillin (Gibco), 100 μg/mL streptomycin (Gibco), 1 mM minimal essential medium non-essential amino acids (Gibco). Cells were incubated at 37 °C in a humidified atmosphere containing 5% CO_2_.

### 2.3. Hybridoma Technology

Hybridoma cells were produced as previously described with slight modifications [31,32]. After immunization with BMDCs, the mice were euthanized after three additional boosters. Antibody-producing B lymphocytes were isolated from the spleens of immunized mice. Cell fusion was performed between the antibody-producing B lymphocytes and P3 × 63-Ag8-653 murine myeloma cells using polyethylene glycol 1500 (Roche, Basel, Switzerland). After cell fusion, hypoxanthine–aminopterin–thymidine (Sigma-Aldrich, St. Louis, MO, USA) selection was performed, and hypoxanthine–thymidine (Sigma-Aldrich, St. Louis, MO, USA) was subsequently supplemented. The mAbs produced by the hybridomas were obtained from culture supernatants. A Pierce Rapid ELISA Mouse mAb Isotyping Kit (Invitrogen, Waltham, MA, USA) was used to identify the immunoglobulin isotype of the mAbs.

### 2.4. Flow Cytometry

The cells were stained with mAbs obtained from the culture supernatant of the hybridomas, including anti-CD1 (76-7-4), anti-CD172a (pan-myeloid marker SWC3; 74-22-15), anti-CD4 (74-12-4), anti-CD8a (76-2-11), anti-CD16 (G7), and anti-MHC II (MSA3) [33]. An FITC- or APC-conjugated goat anti-mouse IgG antibody (BioLegend, San Diego, CA, USA) was used to label the cells. Flow cytometry was conducted with a BD FACSCalibur instrument (BD Biosciences, Franklin Lakes, NJ, USA). Data analysis was conducted using FlowJo 10.10.0 software (BD Biosciences).

### 2.5. Immunostaining

After fixation in 4% paraformaldehyde in Dulbecco’s phosphate-buffered saline, the cells were stained with cultures derived from the following hybridoma clones PD9-9 and G7 mAbs (anti-CD16). An FITC-conjugated goat anti-mouse IgG antibody (BioLegend) was used to label the cells. Nuclei were stained with TO-PRO-3 iodide (Life Technologies, Carlsbad, CA, USA). Confocal laser microscopy (Carl Zeiss, Oberkochen, Germany) was used to acquire the images.

### 2.6. Cell Proliferation Assay

The cells were labeled with carboxyfluorescein succinimidyl ester (CFSE; Invitrogen) before incubation with the mAb obtained from the culture supernatant of the PD9-9 hybridoma. Concentration-dependent supplementation of PD9-9 culture supernatant was performed. The cells were collected for analysis after five days of incubation at 37 °C in a humidified atmosphere containing 5% CO_2_. Cell proliferation was assessed using flow cytometry with a BD FACSCalibur instrument (BD Biosciences). Data analysis was conducted using FlowJo 10.10.0 software (BD Biosciences).

## 3. Results

### 3.1. Production of mAbs against Porcine DCs

DCs used as immunogens were differentiated from BMCs to develop mAbs against porcine DCs (Figure 1). mAb-producing hybridomas were obtained by immunizing mice with BMDCs as antigens. Antibodies recognizing porcine DCs were identified within the hybridoma clones. Then, antibodies were screened for reactivity with peripheral blood mononuclear cells. It was subsequently decided to exclude mAbs that exhibited non-specific reactivity against other leukocytes. mAbs that were specifically reactive with porcine BMDCs were chosen, and their isotypes were classified. Each antibody comprised a heavy immunoglobulin G and a light kappa chain (Table 1). The PD9-9 clone was selected and expanded for subsequent mAb screening because of its remarkable capacity to produce mAbs with extremely high reactivity (Appendix A).

PD9-9 mAbs identified porcine DC populations that expressed CD16 and CD1 and had high MHC II expression levels (Figure 2a). PD9-9 mAb bound to porcine DCs, consistent with anti-porcine CD16 (G7 mAb) (Figure 2b). In summary, this study identified the PD9-9 mAb that recognizes porcine DCs. Furthermore, its applicability in flow cytometry and immunocytochemistry was confirmed.

### 3.2. mAb PD9-9 Reactivity during DC Differentiation

The reactivity of the PD9-9 mAb to cells was assessed during the in vitro differentiation of BMCs to DCs. mAb reactivity kinetics during DC differentiation were ascertained through comparisons with MHC class II, CD16, CD1, and CD172a molecules (Figure 3). PD9-9 mAb-detectable cells were observed on day six of differentiation, and approximately 73.6% of these cells were positive (Figure 3c). In contrast to MHC II expression, which exhibited two distinct populations distinguished by high and low expressions by day 10, PD9-9 mAbs exhibited consistently high reactivity levels in 95.7% of cells (Figure 3d).

### 3.3. Distinguishing DCs from Macrophages

DCs and macrophages share numerous characteristics; therefore, the presence of shared markers may make it challenging to distinguish between the two. We examined the reactivity of PD9-9 antibodies towards porcine macrophages. PD9-9 mAbs did not recognize the porcine alveolar macrophage cell line 3D4/2 (Figure 4a). Additionally, primary macrophages isolated from the pulmonary alveoli demonstrated minimal reactivity (Figure 4b). In summary, despite its high reactivity towards DCs, PD9-9 exhibited minimal reactivity towards macrophages. This finding suggests that PD9-9 mAbs could distinguish between DCs and macrophages.

### 3.4. Effects on DC Proliferation

Certain mAbs modulate or control cellular mechanisms via antigen recognition [34,35]. This study investigated the effect of PD9-9 mAbs on DC proliferation to ascertain their functionality. After dose-dependent treatment with PD9-9 mAbs, DC proliferation was assessed using the CFSE assay. Administering PD9-9 mAb resulted in a dose-dependent increase in DC proliferation in the range of 41.6–64.7% (Figure 5).

## 4. Discussion

DCs migrate to lymphoid organs, where they present antigens to T cells [36,37]. DCs play a crucial role in priming naïve T cells, thereby initiating an adaptive immune response [38,39]. Thus, DCs are an essential bridge between innate and adaptive immunity during pathogen infection [40,41]. The capacity of DCs to induce antigen-specific immune responses has been employed to develop cell therapy technologies [42,43]. This study developed a mAb specific to porcine DCs, thereby contributing to the advancement of knowledge regarding the biology of porcine DCs. In most studies investigating DCs, monocytes or BMCs were cultured with GM-CSF, with or without IL-4, to induce DC differentiation in vitro. In this study, one of the developed mAbs exhibited remarkable reactivity towards porcine DCs and was designated PD9-9. The PD9-9 mAb application included flow cytometry and immunostaining. Moreover, the mAbs exhibited no reactivity towards porcine macrophages. Distinguishing DCs from macrophages has proven challenging because of the similarity of their properties [44,45]. Hence, the PD9-9 mAb may be a useful tool for distinguishing porcine DCs from macrophages.

Before differentiation, the PD9-9 antibody did not recognize BMCs; however, it exhibited cell reactivity on day 6 after inducing DC differentiation. In addition to being applicable to fully differentiated porcine DCs, the PD9-9 antibody could identify DCs in the process of differentiation. In a previous study, porcine BMDCs were differentiated with GM-CSF on day 10 to distinguish between immature (MHCII^low^) and mature (MHCII^high^) DCs based on differential MHC II expression [30]. MHCII^high^ and MHCII^low^ cells were both possessed the capacity to prime T cells, a signature function of DCs. All BMDCs, including immature DCs with low MHCII expression, were identified by PD9-9 on day 10. PD9-9 was also capable of identifying immature DCs that emerged on day 6 and were characterized by MHCII^low^ cells. Because PD9-9 mAbs recognized immature and mature BMDCs, they can be used as a biomarker for porcine BMDCs and in the study of DC ontogeny.

Antibodies are essential components of the immune system that function by circulating within the bloodstream and identifying exogenous substances [46,47]. The emergence of hybridoma technology, which enables manufacturing mAbs, has driven biotechnology forwards [48,49]. This study generated PD9-9 mAb against porcine BMDCs differentiated with GM-CSF using hybridoma technology. PD9-9 mAbs could identify porcine DCs that had completed differentiation and cells in the process of DC differentiation but were inactive against bone marrow progenitor cells. Antibodies generated using hybridoma technology may possess functionalities in addition to antigen recognition [50]. PD9-9 also has the ability to promote cell proliferation in porcine BMDCs when it is administered. Considering the broad roles of DC in immunobiology, the ability of PD9-9 to promote DC proliferation could potentially serve a therapeutic purpose. The immunocytochemistry results indicate that the PD9-9 mAb reactivates the protein that are located on the surface of BMDCs. Hence, it is probable that the antigen corresponds to one of the cell surface proteins present on porcine DCs. The epitope is the specific region of the antigen that the antibody recognizes, and it can be complex for proteinaceous antigen [51]. The binding epitope of an antibody can offer significant mechanistic insights and suggest potential applications for the antibody. Recently, a variety of advanced methods has been developed and is extensively employed to predict antigens and epitopes [52]. Additional research is required to identify the specific protein antigen and epitope that are targeted by the PD9-9 mAb, using these advanced approaches. Consequently, the PD9-9 mAb developed in this research will be the most effective to utilize.

## 5. Conclusions

In conclusion, PD9-9 mAbs recognized fully differentiated porcine BMDCs and cells that were in the process of undergoing DC differentiation. In order to distinguish between macrophages, the PD9-9 mAb can be employed as an additional porcine DC marker. Additionally, in vitro DC proliferation is promoted by the PD9-9 mAb. Thus, the PD9-9 mAb could be a novel tool for investigating the immune ontogeny of porcine DC cell types and studying the biology of porcine DCs.

## Figures and Tables

**Figure 1 life-14-01054-f001:**
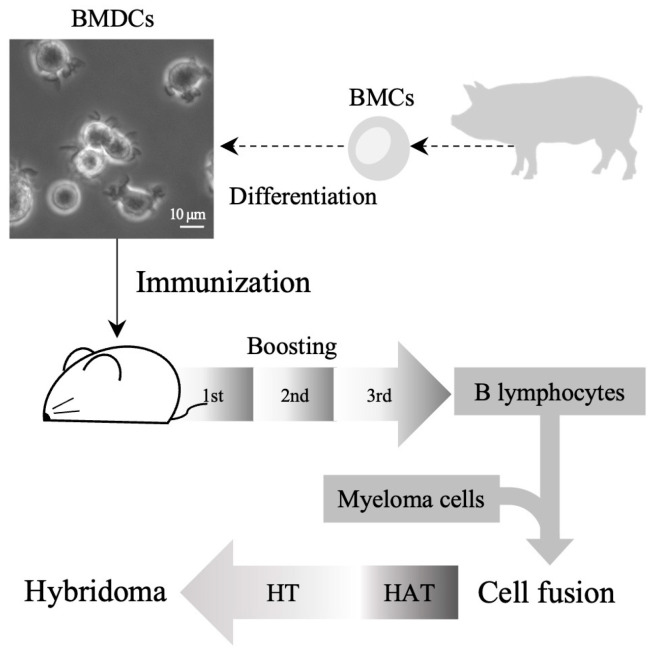
Diagram illustrating the production process of monoclonal antibodies (mAbs) against porcine dendritic cells (DCs). Porcine bone marrow cells (BMCs) were isolated and differentiated into DCs. B lymphocytes from mice were obtained after four immunizations with porcine bone marrow-derived DCs (BMDCs). Cell fusion was performed between murine B lymphocytes and myeloma cells. Antibody-producing hybridomas were obtained via hypoxanthine–aminopterin–thymidine (HAT) selection and hypoxanthine–thymidine (HT) supplementation.

**Figure 2 life-14-01054-f002:**
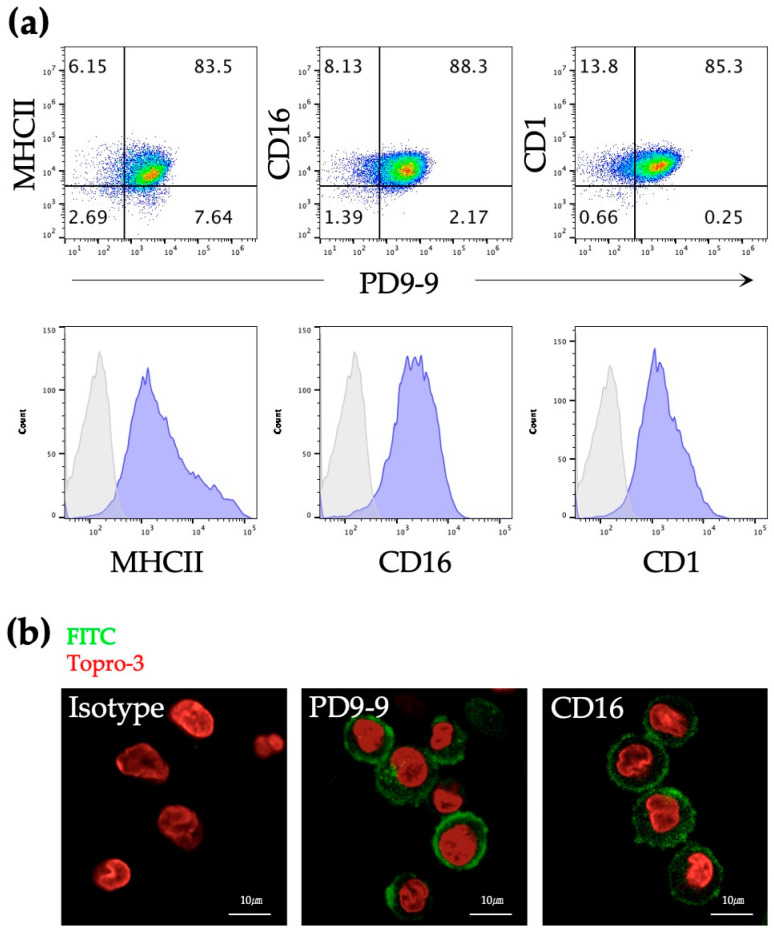
Application of PD9-9 mAbs to identify porcine DCs. (**a**) The identification of porcine DCs stained with PD9-9 mAb in combination with MHC II, CD16, and CD1 was confirmed using flow cytometry. Blue-filled histograms indicate cells that are stained with each mAb, whereas gray-filled histograms represent isotype controls. (**b**) The binding of PD9-9 mAb to porcine DCs is confirmed using immunocytochemistry. The FITC-labeled antibodies that recognize the antigens are denoted in green. Topro-3-stained nuclei exhibit a red luminescence.

**Figure 3 life-14-01054-f003:**
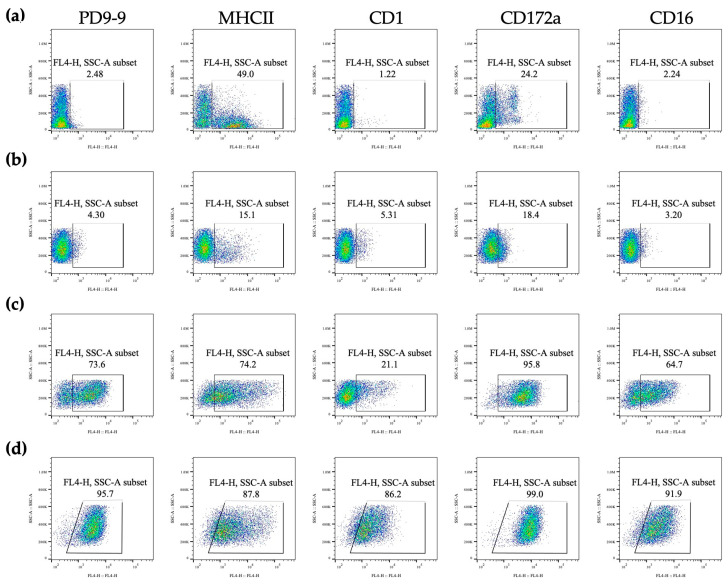
PD9-9 mAb reactivity to cells during BMDC differentiation. Cells were harvested and examined on days 0 (**a**), 3 (**b**), 6 (**c**), and 10 (**d**) using flow cytometry during BMDC differentiation. Cells were stained with the indicated mAb.

**Figure 4 life-14-01054-f004:**
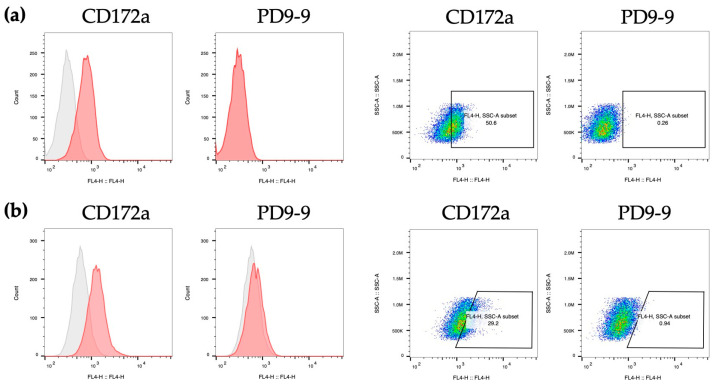
Evaluation of PD9-9 mAb reactivity towards porcine macrophages. (**a**) The reactivity of PD9-9 mAbs with 3D4/2 cells is evaluated. (**b**) The reactivity of PD9-9 mAb against pulmonary alveolar macrophages is evaluated. Red histograms represent cells stained with the indicated antibody, whereas gray histograms denote the isotype control.

**Figure 5 life-14-01054-f005:**
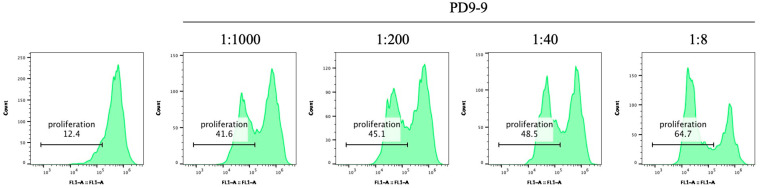
The application of PD9-9 mAbs to promote porcine DC proliferation. Porcine DCs are labeled with the carboxyfluorescein succinimidyl ester (CFSE) and incubated for 5 days in the presence of PD9-9 mAbs in a ranging dose. Cell proliferation is verified using the CFSE assay and assessed via flow cytometry.

**Table 1 life-14-01054-t001:** Isotype of monoclonal antibodies against porcine dendritic cells.

Clone	Heavy Chain	Light Chain
PD9-7	IgG1	ϰ
PD9-9	IgG1	ϰ
PD10-3	IgG1	ϰ
G7	IgG1	ϰ

## Data Availability

Data are contained within the article or the Appendix A.

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
