# Peer review of "Development of the PD9-9 Monoclonal Antibody for Identifying Porcine Bone Marrow-Derived Dendritic Cells"

_life, 2024, doi:10.3390/life14091054_

Round 1

Reviewer 1 Report

Comments and Suggestions for Authors

This manuscript describes the production of monoclonal antibodies (mAb) specific to porcine bone marrow-derived dendritic cells (BMDCs). The animal use and care protocol was approved by the IACUC. The methods and results are clearly explained. However, it would be clearer if:

 1. Are there any ideas, opinions, or evidence to indicate the target of the P9-9 mAb?

 2. There is a possible typo on line 203, "immature DCs".

Author Response

Thank you very much for taking the time to review this manuscript.

We fully considered your comments and we revised your all valuable comments as we can.

Please find the detailed responses below and the corresponding revisions/corrections highlighted/in track changes in the re-submitted files.

Comments 1: Are there any ideas, opinions, or evidence to indicate the target of the P9-9 mAb?

Responses 1: The immunocytochemistry data suggest PD9-9 mAb recognizes proteins on the surface of DCs (Figure 2b). Unfortunately, our attempts to identify protein antigens through immunoprecipitation and Western blot weren't successful. We consider that PD9-9 mAb binds to conformational epitopes rather than linear epitopes. We have been trying to identify the protein antigen and epitope of PD9-9 mAb through advanced approaches.

Comments 2: There is a possible typo on line 203, "immature DCs".

Responses 2: Thank you for noticing the typo. Corrected it for “immature DCs” (line 207).

Reviewer 2 Report

Comments and Suggestions for Authors

                The authors have endeavored to create a monoclonal antibody that recognizes porcine DCs. They have provided arguments indicating that such an antibody would be useful, given the importance of pigs to agriculture and medicine, and their risk of viral infection. An antibody was developed, named PD9-9, that appears to react only with BM-derived porcine cells that have undergone standard DC differentiation with GM-CSF, and display other known markers of DCs. The manuscript would be improved by attention to the following concerns:

                Major criticism:

The authors have not met a minimum standard of validation for their antibody. At least one of the following should be done:

1) Establish that cells identified by PD9-9, to a high proportion, perform functionally as DCs able to activate MHC-matched T cells. There are many assays that could be used for this purpose, both in vitro or in vivo.

2) Perform more elaborate characterization supporting that PD9-9 is a marker of DCs. This could come from demonstrating upregulation of co-stimulatory markers and Signal 3 cytokines in response to conditioning stimuli such as IFN-gamma or TLR agonists.

3) Attempt to identify the antigen recognized by PD9-9, and/or show that it is not simply one of the well-known markers of DCs. A Western blot with PD9-9, followed by mass spectrometry, would be the standard approach.

                Minor criticism:

The authors should explain why stillborn piglets were the source of BM for the generation of the antibody. Furthermore, they should clarify whether BM from stillborn piglets was the only source of cells for antibody testing; if it was, then PD9-9 should be tested against cells derived from live adult pigs.

Author Response

Thank you very much for taking the time to review this manuscript. We fully considered your comments and we revised your all valuable comments as we can. Please find the detailed responses below and the corresponding revisions/corrections highlighted/in track changes in the re-submitted files.

-------------------------------------------------------------------

Major criticism:

The authors have not met a minimum standard of validation for their antibody. At least one of the following should be done:

Comments 1: Establish that cells identified by PD9-9, to a high proportion, perform functionally as DCs able to activate MHC-matched T cells. There are many assays that could be used for this purpose, both in vitro or in vivo.

Response 1: We agree with this comment. In our previous study, we characterized BMDCs that was induced to differentiate with GMCSF (ref.30). In that study, GMCSF-BMDCs were capable of priming T cells, as demonstrated by a mixed lymphocyte reaction assay. We considered it necessary to address the functional functions of dendritic cells, as you pointed out. Thus, we incorporated this information into the discussion section on lines 204-205 (page 6).

Comments 2: Perform more elaborate characterization supporting that PD9-9 is a marker of DCs. This could come from demonstrating upregulation of co-stimulatory markers and Signal 3 cytokines in response to conditioning stimuli such as IFN-gamma or TLR agonists.

Response 2: We appreciate your improvement suggestions. A further study is planned to determine the antigen that the PD9-9 mAb targets. We will evaluate the responsiveness of PD9-9 mAb by stimulating DCs and analyzing the upregulation of co-stimulatory markers and cytokines, as per your recommendation.

Comments 3: Attempt to identify the antigen recognized by PD9-9, and/or show that it is not simply one of the well-known markers of DCs. A Western blot with PD9-9, followed by mass spectrometry, would be the standard approach.

Response 3: We agree with this comment. To identify the antigen, we conducted western blot and immunoprecipitation. Regrettably, the protein antigen was not detected. We consider that PD9-9 mAb binds to conformational epitopes rather than linear epitopes. We have been trying to identify the protein antigen and epitope of PD9-9 mAb through advanced approaches.

Minor criticism:

Comments 4: The authors should explain why stillborn piglets were the source of BM for the generation of the antibody. Furthermore, they should clarify whether BM from stillborn piglets was the only source of cells for antibody testing; if it was, then PD9-9 should be tested against cells derived from live adult pigs.

Response 4: Stillborn piglets were employed in this study to reduce the scope of unnecessary animal sacrifice. BMDCs from living 3-week-old piglets from our previous study were employed to verify the response of PD9-9 mAb. PD9-9 mAb also identified DCs that had been differentiated from bone marrow cells that were isolated from living piglets.

Reviewer 3 Report

Comments and Suggestions for Authors

Kim et al. report in this paper the production and characterization of a monoclonal antibody PD9-9 which was produced by immunizing mice with porcine bone marrow-derived dendritic cells (DCs). The data presented in this paper show clear reactivity of PD9-9 with DCs and thus could be an interesting tool to study DCs in pigs. In particular, the finding that PD9-9 is enhancing the proliferation of DCs is highly interesting.

The following questions and suggestions were raised during the review process:

1)     How is maturation of porcine DCs (e.g. induced by LPS etc) regulating the expression/reactivity of mAb PD9-9?

2)     Is PD9-9 reacting with DCs in skin or other tissues? Does mAb PD9-9 recognize DCs in the blood of pigs?

3)     The authors show that PD9-9 reacts with DCs but not with macrophages. Yet, it would be important to show if PD9-9 is or is not reacting with other leukocytes, eg. PBMNCs in the blood? This information would be very important to judge the specificity of the reactivity.

4)     Do the authors have any information about the molecular weight or sequence information about the cell surface molecule to which PD9-9 is binding?

5)     The induction of T cell proliferation is a key function of DCs. Is mAb PD9-9 influencing this function? This could be tested for instance in an MLR.

6)     Does PD9-9 alter the phagocytic capacity of DCs?

7)     The authors show in Figure 5 that PD9-9 is enhancing the proliferation of DCs. Yet, the data could be also interpreted that PD9-9 is inducing proliferation, since untreated cells seem not to proliferate at all. To my knowledge, DCs are considered as non-proliferating cells and there is no trigger reported which is able to induce proliferation of primary DCs. So, this is a critical point to distinguish between “enhancing” or “inducing” proliferation. Was the proliferation assay done in the presence or absence of GM-CSF?

8)     The Discussion is very short and does not contain major aspects, such as the potential candidates for the PD9-9-defined antigen, the function and specificity of the reactivity.

Author Response

Thank you very much for taking the time to review this manuscript. We fully considered your comments and we revised your all valuable comments as we can. Please find the detailed responses below and the corresponding revisions/corrections highlighted/in track changes in the re-submitted files.

Comments 1: How is maturation of porcine DCs (e.g. induced by LPS etc) regulating the expression/reactivity of mAb PD9-9?

Response 1: In our previous study, porcine BMDCs were differentiated with GM-CSF on day 10 to distinguish between immature (MHCIIlow) and mature (MHCIIhigh) DCs [ref. 30]. The PD9-9 mAb was capable of recognizing both mature and immature DCs. Therefore, it is anticipated that the reactivity of PD9-9 will be preserved as DCs mature in response to a variety of stimuli. We would like to do a follow-up study to examine the disparity in the responsiveness of antibodies to DCs that have been triggered by various stimuli, as you have recommended.

Comments 2: Is PD9-9 reacting with DCs in skin or other tissues? Does mAb PD9-9 recognize DCs in the blood of pigs?

Response 2: We appreciate your improvement suggestions. We would like to do a follow-up study to identify the reactivity of DCs in skin and lymphoid tissues by immunohistochemistry. The reactivity of dendritic cells in the blood was not able to be tested due to the difficulty of isolating them, as the number of these cells is exceedingly small.

Comments 3: The authors show that PD9-9 reacts with DCs but not with macrophages. Yet, it would be important to show if PD9-9 is or is not reacting with other leukocytes, eg. PBMNCs in the blood? This information would be very important to judge the specificity of the reactivity.

Response 3: We agree with this comment. We initially selected clones that reacted with BMDCs in order to develop antibodies against porcine DCs. Subsequently, we screened for reactivity against PBMCs to rule out antibodies with non-specific reactivity to other immune cells. Additionally, as you noted, it is crucial to assess reactivity against other leukocytes. Thus, we considered it important to include this information in the results section, on lines 118-122 (pages 3-4). Thank you for pointing out what I missed.

Comments 4: Do the authors have any information about the molecular weight or sequence information about the cell surface molecule to which PD9-9 is binding?

Response 4: The immunocytochemistry data suggest PD9-9 mAb recognizes proteins on the surface of DCs (Figure 2b). Unfortunately, our attempts to identify protein antigens through immunoprecipitation and Western blot weren't successful. We consider that PD9-9 mAb binds to conformational epitopes rather than linear epitopes. We have been trying to identify the protein antigen and epitope of PD9-9 mAb through advanced approaches.

Comments 5: The induction of T cell proliferation is a key function of DCs. Is mAb PD9-9 influencing this function? This could be tested for instance in an MLR.

Response 5: In our previous study, we characterized BMDCs that was induced to differentiate with GMCSF (ref.30). In that study, GMCSF-BMDCs were capable of priming T cells, as demonstrated by a mixed lymphocyte reaction assay. As per your suggestion, we would like to do a follow-up study to validate if this antibody affects the T cell priming function of DCs.

Comments 6: Does PD9-9 alter the phagocytic capacity of DCs?

Response 6: The phagocytic capacity of GMCSF-BMDCs was also identified in our previous work (ref.30). In accordance with your recommendation, we would like to conduct a follow-up study to confirm whether this antibody influences the phagocytic capacity of DCs.

Comments 7: The authors show in Figure 5 that PD9-9 is enhancing the proliferation of DCs. Yet, the data could be also interpreted that PD9-9 is inducing proliferation, since untreated cells seem not to proliferate at all. To my knowledge, DCs are considered as non-proliferating cells and there is no trigger reported which is able to induce proliferation of primary DCs. So, this is a critical point to distinguish between “enhancing” or “inducing” proliferation. Was the proliferation assay done in the presence or absence of GM-CSF?

Response 7: We agree with this comment. As you stated, dendritic cells are cells are non-proliferating cells. The proliferation assay was conducted in the absence of GM-CSF. Thus, we have replaced the term "enhance" with "promote" in both the abstract (line 19 on page 1) and discussion (line 218 on page 7) sections.

Comments 8: The Discussion is very short and does not contain major aspects, such as the potential candidates for the PD9-9-defined antigen, the function and specificity of the reactivity.

Response 8: As per your valuable comments, we added to the discussion. As you correctly emphasize, it is essential to determine the antigen of PD9-9, and we have provided this information in the discussion from lines 220 to 230 (page 7).

Round 2

Reviewer 2 Report

Comments and Suggestions for Authors

The authors have provided adequate responses to my criticisms.

Comments on the Quality of English Language

Minor editing for grammar is recommended.

Reviewer 3 Report

Comments and Suggestions for Authors

The authors have addressed all suggestion, questions and comments which were raised during the review process. So, the paper is now suitable for publication.